# Exploring Amino Acid Transporters as Therapeutic Targets for Cancer: An Examination of Inhibitor Structures, Selectivity Issues, and Discovery Approaches

**DOI:** 10.3390/pharmaceutics16020197

**Published:** 2024-01-30

**Authors:** Sebastian Jakobsen, Carsten Uhd Nielsen

**Affiliations:** Department of Physics, Chemistry and Pharmacy, University of Southern Denmark, Campusvej 55, DK-5230 Odense, Denmark

**Keywords:** inhibitors, amino acids, solute carriers, cancer, ASCT2, LAT1, xCT, SNAT1, SNAT2, PAT1

## Abstract

Amino acid transporters are abundant amongst the solute carrier family and have an important role in facilitating the transfer of amino acids across cell membranes. Because of their impact on cell nutrient distribution, they also appear to have an important role in the growth and development of cancer. Naturally, this has made amino acid transporters a novel target of interest for the development of new anticancer drugs. Many attempts have been made to develop inhibitors of amino acid transporters to slow down cancer cell growth, and some have even reached clinical trials. The purpose of this review is to help organize the available information on the efforts to discover amino acid transporter inhibitors by focusing on the amino acid transporters ASCT2 (SLC1A5), LAT1 (SLC7A5), xCT (SLC7A11), SNAT1 (SLC38A1), SNAT2 (SLC38A2), and PAT1 (SLC36A1). We discuss the function of the transporters, their implication in cancer, their known inhibitors, issues regarding selective inhibitors, and the efforts and strategies of discovering inhibitors. The goal is to encourage researchers to continue the search and development within the field of cancer treatment research targeting amino acid transporters.

## 1. Introduction

The Solute Carrier (SLC) gene family encodes structurally diverse groups of membrane proteins that facilitate the transport of nutrients, metabolites, vitamins, and minerals, as well as several small molecular drug substances across cell membranes. Solute carriers are the second largest family of membrane proteins, with a current number of 448 members across 66 different subfamilies [1]. Despite their implications in various diseases, SLCs are surprisingly unexplored as pharmacological targets compared to more popular targets such as ion channels, G protein-coupled receptors, and kinases [2,3]. One disease where SLCs have gained some interest is cancer, where they function as facilitators of cancer cell growth or, on the contrary, as tumor cell suppressors [4,5,6,7,8]. Due to their high proliferation, cancer cells need substantial amounts of nutrients to fuel their energetic and biosynthetic demands, making SLCs important metabolic gatekeepers. In the 1920s, Warburg and colleagues discovered that cancer cells take up immense amounts of glucose when compared to cells at rest [9]. The high glucose consumption was used for glycolysis as well as lactic acid fermentation, even in the presence of oxygen, coining the terms “aerobic glycolysis” and the “Warburg effect”. This uptake of glucose is mainly facilitated by the increased expression of GLUT1 (SLC2A1) in various cancer cells [10]. Hence, GLUT1 is central in the diagnosis and detection of tumors, where the radiolabeled glucose analog ^18^F-fluorodeoxyglucose (FDG), which is a substrate of GLUT1, is used to detect and quantify the glucose utilization of tumors [11]. Other SLCs have also been shown to be useful in cancer detection. For instance, the sodium/iodide symporter (NIS, SLC5A5) is responsible for the uptake of radioisotopes of iodide in thyroid cancers [12]. It has also been shown that ^11^C-sarcosine might have a use as a new PET imaging probe due to its accumulation in prostate cancer tissue [13]. This uptake of sarcosine was first believed to be mainly PAT- (SLC36) mediated, but it appears that SNAT2 (SLC38A2) also plays an important role in sarcosine uptake in prostate cancer cells, at least in vitro [14].

The human genome encodes at least 60 SLCs that can transport amino acids, found across 11 different SLC subfamilies. Some of these have also been shown to have an importance in the metabolism in cancer cells, given the role of amino acids in protein synthesis, activation of the cell growth regulator mTORC1 (mammalian target of rapamycin complex 1), and various other metabolic pathways (as reviewed in [15]). In particular, the role of glutamine transporters has been examined. Where the Warburg effect is a term for the rewired glucose metabolism, the term “glutamine addiction” points to the increased glutamine consumption of cancer cells [16,17]. Glutamine has an important role in providing its γ-nitrogen for nucleotide synthesis, converting it to glutamic acid, which in turn can donate its α-nitrogen to the synthesis of non-essential amino acids [18]. Furthermore, glutamine can be used to replenish the tricarboxylic acid (TCA) cycle with metabolites, which in some cancer cells is sparsely supplied by glucose-derived pyruvate due to the Warburg effect. This anaplerotic reaction of glutamine is known as glutaminolysis, where glutamine is first converted into glutamate and then into α-ketoglutarate, which is a substrate of the TCA cycle. Glutaminolysis can then provide cancer cells with critical precursors for synthesizing non-essential amino acids, nucleotides, and lipids (as reviewed in [19]). It is also believed that glutamine has an important role in being an exchange substrate for amino acid antiporters, such as LAT1 (SLC7A5), where it is effluxed to drive the uptake of essential amino acids such as leucine [20]. This, of course, has made glutamine metabolism and uptake an interesting possible target in the discovery of anticancer drugs. Various SLCs can transport glutamine, but specifically, ASCT2 (SLC1A5) has gained interest as a target for inhibiting glutamine uptake, given its general overexpression in tumor tissue and its use as a prognostic biomarker [21]. However, as pointed out by Bröer and colleagues, ASCT2 is an amino acid exchanger and, therefore, needs to trade the uptake of glutamine with the efflux of another amino acid substrate [22]. The SLC38 transporters SNAT1 (SLC38A1) and SNAT2 (SLC38A2) are symporters that utilize a Na^+^ gradient to drive the uptake of glutamine as well as other amino acids in cells. Bröer et al. refer to these amino acid transporters as “loaders” because they facilitate the net uptake of amino acids by coupling to an ion gradient. On the other hand, transporters like LAT1 and ASCT2 are called “harmonizers” because of their role as exchangers to balance out the amino acid pool within the cell by trading the accumulated “loader” substrates like glutamine and alanine for other essential amino acids [22]. This review focuses on the amino acid transporters ASCT2, LAT1, xCT, SNAT1, SNAT2, and PAT1, which all have been implicated in cancer and spans well-explored (ASCT2, LAT1, xCT) and less explored (SNAT1, SNAT2, PAT1) SLCs in terms of structural biology and inhibitor identification. The interconnection between these amino acid transporters and their role in cancer can be seen in Figure 1. Overall, this exemplifies the important role of amino acid transporters in the rewired metabolism of cancer cells. Therefore, modulation of amino acid transport activity, e.g., in the form of inhibitors of the translocation cycle, appears as a potential pharmacological class of novel anticancer agents. The present review aims to provide a broad state-of-the-art overview of amino acid transporters implicated in cancer, their known substrates, and inhibitors, as well as discuss the different methodologies used to discover such inhibitors and how new ones can be uncovered. Previous reviews on this topic have focused on their structural aspects and their physiological/pathological functions [15,23,24], as well as provided lists of cancer-relevant amino acid transporters and their most common inhibitors, along with inhibitor selectivity [25]. This review differs by highlighting the chemical structures of these inhibitors, discussing the struggles in designing selective inhibitors, and presenting discovery approaches that together should inspire researchers to make discoveries within this field.

## 2. Discovering Amino Acid Transporter Inhibitors

When studying molecules that inhibit SLCs, it is important to be familiar with key transporter terminology [26]. Albeit that most researchers interested in this field are acquainted with these concepts, we want to provide a section that helps researchers new to the field become familiar with important terms and concepts. The movement of a substrate across a membrane, where the transporter moves from the outward open to the inward open conformation, is called a translocation cycle. The transporter turnover rate is the number of molecules moved by a single transporter per second [27]. Substrates are compounds that bind to and are translocated by the solute carrier. Substrates are generally characterized by their Michaelis constant (K_m_), which reflects the substrate concentration where half of the maximal transport velocity is achieved. However, when two substrates compete for binding, one can be considered a “competitive inhibitor” of the translocation of the substrate in question. This is due to competitive binding to the substrate binding pocket of the transporter. Calling such substrates inhibitors might be misleading as it only indicates that the substrate can inhibit the translocation of other substrates. True inhibitors are molecules that bind to the transporter without being translocated themselves. This can be due to competitive inhibition at the substrate binding site. Binding to this site is called orthosteric, whereas allosteric refers to a site other than the substrate binding site. Un-competitive inhibition is when the inhibitor only binds the substrate-transporter complex. Non-competitive binding is when the inhibitor equally binds the free transporter and substrate-transporter complex. If the inhibitor binds the free transporter and the substrate-transporter complex with unequal affinity, this is called mixed-inhibition. Another commonly used term is ‘ligand’, which generally refers to any small molecule that is able to bind to a macromolecule like a transporter. Whether the molecule is translocated or not is often not addressed when the term ligand is used. To compare inhibitors, affinity or potency are used, which are terms that are generally more defined in receptor pharmacology. Affinity reflects the binding of the inhibitor to the transporter and is usually quantified through the dissociation constant (K_d_), which is the inhibitor concentration where 50% of the transporters are bound by the inhibitor. Potency reflects the ability of the compound to inhibit the transporter function and is usually quantified by the IC_50_, which is the inhibitor concentration that leads to 50% inhibition of the transport activity. The IC_50_ is highly dependent on the substrate concentration used in the assay to determine this value. An absolute value that is easier to compare is the inhibition constant (K_i_), which is independent of the substrate concentration. The relation between the IC_50_ and the K_i_ is dependent on the different modes of inhibition (i.e., competitive, un-competitive, non-competitive, and mixed), which is seen in the so-called Cheng-Prusoff equations [28]. Generally, the binding of the inhibitor to the transporter is also indicative of its ability to inhibit, which is why affinity is sometimes used synonymously with potency. Selectivity refers to the ability of the compound to selectively bind to a given transporter without binding to other related transporters. Often, selectivity is a matter of differences in affinity relative to the concentration of inhibitor the transporter is exposed to, as compounds tend to bind to multiple targets at higher concentrations. For instance, the selective serotonin uptake inhibitor sertraline binds the serotonin transporter (SERT, SLC6A4) with a K_d_ of 0.29 nM but also binds the norepinephrine transporter (NET, SLC6A2) with a 1400 times lower affinity (K_d_ = 420 nM) [29]. So, the typical goal in the development of SLC inhibitors is to find a compound that is non-translocated, has great inhibitory potency, and is selective to limit any unwanted effects due to binding to other targets. It is generally more difficult to find transporter inhibitors with similarly low potencies as compounds targeting receptors, which makes potencies in the submicromolar range a great accomplishment in the transporter field. Since some inhibitors might work by inhibiting the transporter at an intracellular site, the inhibitor’s ability to cross the plasma membrane by preferably passive diffusion is an important property of transporter inhibitors. Inhibitors with greater lipophilicity have higher permeabilities across lipid barriers but at the detriment of aqueous solubility. Higher lipophilicity can lead to an increased potency since lipophilic compounds prefer to stick to protein cavities rather than being fully exposed to water. However, lipophilicity is also commonly associated with decreased selectivity, increased toxicity, increased protein binding, and increased metabolic clearance (reviewed in [30]). So, in addition to having a high potency and selectivity, the ideal inhibitor should have sufficient solubility and oral bioavailability (>85%), low metabolism and protein binding, and be cleared really by glomerular filtration without any secretion or re-absorption. 

Several strategies and methodologies can be pursued to discover SLC inhibitors. The more traditional approach is through high-throughput screening (HTS). Here, a large library of compounds is screened for their inhibition using various functional in vitro assays, like an uptake assay of a radiolabeled or fluorescent substrate. A nice overview of available cell-based assays for the study of SLCs can be found in the following review [31]. Of course, this approach requires a large library of chemically diverse compounds as well as a scalable assay that can screen many compounds at a time. An example of this approach has recently been used to discover inhibitors of SNAT2 by screening a library of 33,934 compounds [32]. However, a more focused screen can be implemented if resources are limited or if only low-throughput assays are available. This approach usually screens structural analogs of already known substrates or inhibitors in the pursuit of more potent ligands. Another and more modern way to narrow down the search is to implement in silico methods. Broadly speaking, computer-aided drug design (CADD) can be categorized into two main categories: ligand-based drug design and structure-based drug design ([33] for review). The ligand-based approach uses known ligands to generate a model that can help predict novel ligands. In its simplest form, this can be a similarity search that finds chemically similar molecules to known ligands. More sophisticated is a pharmacophore model, which is a spatial representation of the key chemical features of a ligand and its possible interactions with its biological target. The pharmacophore model can then be used to query virtual compound libraries to find other compounds that fit the pharmacophore model and are possible ligands. A ligand-based approach has been used to study the human proton-coupled amino acid transporter 1 (hPAT1) and helped identify a novel scaffold for hPAT1 substrates [34]. Structure-based drug design, on the other hand, focuses on the structure and binding site of the biological target and thus relies on a solved 3D structure of the protein (e.g., through X-ray crystallography or cryo-EM). If no structure is available, homology modeling can be used, where a homologous protein structure is used as a template to model the protein of interest. Using the protein structure, a virtual compound library can be screened by docking molecules into the binding site. The resulting docking score can then be used to predict which compounds are likely to be ligands. This often requires a protein structure that includes a ligand to define the binding site; otherwise, this needs to be defined through other means. Virtual screening through molecular docking is known to be most effective when screening ultra-large compound libraries [35]. To make docking of larger libraries more efficient, machine learning can be included to guide the docking, which has recently been demonstrated in the discovery of P-glycoprotein inhibitors [36]. Structure-based drug design can also implement molecular dynamic simulations that simulate both the movement of the protein and ligand, contrasting docking that usually treats the protein as being rigid. One instance of structure-based drug design for amino acid transporters can be seen for LAT1, where docking to a homology model was used to find possible inhibitors that could be tested experimentally [37]. It is uncommon to use only one of these approaches exclusively, as they often work best in collaboration. For instance, in silico methods can be used to generate a more focused screen, and structure-based drug design can also be used to generate pharmacophore models that can be compared to ligand-based ones.

However, there are some obvious concerns and limitations that must be taken into consideration that make it more difficult to develop pharmacological agents that inhibit SLCs when compared to other protein targets. For most SLCs and also for AA transporters, limited knowledge about their ligands or other compounds that might interact with these proteins is available; some are even described as “Orphan” meaning that the endogenous substrate is not known. This limits the chemical tools available for studying these proteins [2]. Moreover, there is a lack of structural information on SLCs due to the difficulty in crystallizing membrane proteins. However, in recent years, several SLCs have been deorphanized, and there has been an increase in the amount of available SLC crystal structures, several of which are amino acid transporters [38]. The recent development of AI has also allowed AlphaFold2 to accurately predict the structure of proteins that lack crystal structures, which can be very helpful for structure-based drug design against SLCs [39]. So, this is a prime age for exploring amino acid transporters and SLCs in general as pharmacological targets and helping expand our knowledge of their function, structure, and chemical probes. The following sections will describe some notable amino acid transporters implicated in cancer and the efforts to develop inhibitors against these SLCs. A list of available protein database (PDB) structures of the discussed amino acid transporters can be seen in Table 1.

## 3. Alanine, Serine, Cysteine Transporter 2 (ASCT2)

### 3.1. Background

In 1967, Christensen and colleagues discovered that Ehrlich cells had a Na^+^-dependent amino acid transport component that preferred the substrates Ala, Ser, and Cys. Before knowing the amino acid transporters responsible for the observed amino acid transport, these transport processes were categorized into so-called transport systems. The above transport system was named system ASC for its substrate preference and to distinguish it from other amino acid transport systems like system L and system A, which preferred Leu and Ala, respectively [40]. A similar system was found in bovine kidney epithelia and named B^0^ for its sodium dependency and broad substrate specificity of neutral amino acids [41]. This resulted in the names ASCT2 and ATB^0^ describing the same system, but in 2000, it was determined to be attributed to a single transporter that has retained the name ASCT2 [42]. The human transporter was first cloned in 1996 by screening a human placental choriocarcinoma cDNA library using the earlier identified *ASCT1 cDNA* as a probe [43,44,45]. ASCT2 was placed in the SLC1 subfamily (SLC1A5) along with the excitatory amino-acid transporters (EAATs) and ASCT1. ASCT2 functions as a Na^+^-dependent amino acid homo- and hetero-exchanger [40,46]. Despite its ASC namesake, ASCT2 has a broader substrate specificity than the related ASCT1 and accepts a variety of neutral amino acids such as Gln, Ala, Thr, Ser, Met, Val, and Leu [46,47]. The Michaelis constant of the preferred substrate Gln has been determined to have a K_m_ in the range of 45–97 µM [46,47,48]. However, its precise transport mechanism in regards to asymmetric antiport, electrogenicity, and stoichiometry has been elusive to pinpoint (as reviewed in [49]). For instance, despite the Na^+^ dependence, the overall transport of ASCT2 is electroneutral, thus questioning the proposed stoichiometric and electric nature of the transporter. Proteoliposomal studies have revealed an asymmetry in the antiport, with Ala, Cys, Val, and Met being inwardly transported, whereas Gln, Ser, Asn, and Thr could be transported bi-directionally [48]. Furthermore, Cys was first believed to be a substrate, but this was derived indirectly from inhibition studies. Later, it was shown that Cys is not a substrate of ASCT2 but a specific modulator that can trigger Gln efflux [49,50]. Currently, its physiological role is believed primarily to be a Gln exchanger that trades Gln for other neutral amino acids in a Na^+^-dependent but electroneutral manner [49]. However, it has now been shown that ASCT2 could mediate an inward Na^+^ flux across proteoliposomes with a 2 Na^+^:1 Gln stoichiometry [51]. ASCT2 consists of 541 amino acids with approximately 61% identity with ASCT1 and around 40% identity with the EAATs [43]. The 3D structure was solved through cryo-EM in 2018, showing a homotrimeric organization of the protein, with each subunit forming eight transmembrane domains, which the authors separated into scaffold and transport domains [52]. Located at the plasma membrane, ASCT2 is broadly expressed with the highest expression in the lung, skeletal muscle, large intestine, kidney, testis, T-cells, brain, and adipose tissue [49]. The broad expression of ASCT2 emphasizes its role as a harmonizing transporter, as proposed by Bröer and colleagues, balancing the pool of amino acids within cells to resemble the physiological levels [22]. The function of ASCT2 as a Gln transporter has been shown to have an important role in the differentiation of naïve T-cells upon immune activation [53]. This highlights the role of ASCT2 in stimulating the mTORC1 pathway, which has an important function in cell growth upon environmental stimuli like the availability of amino acids.

### 3.2. ASCT2 Expression and ROLE in Cancer

ASCT2 is upregulated in various cancers such as hepatocellular carcinoma, non-small cell lung cancer (NSCLC), breast cancer, colorectal cancer, gastric cancer, clear cell renal cell carcinoma, and prostate cancer [54,55,56,57,58,59,60]. The expression level of ASCT2 is also strongly related to cancer survival, making it a prognostic biomarker in tumors [21]. One important role for ASCT2 in the progression of cancer is its function as a high-affinity Gln transporter supplying the Gln-addicted cancer cells as has been demonstrated in NSCLC, prostate cancer, triple-negative breast cancer, head and neck squamous cell carcinoma, and oral squamous cell carcinoma cells [60,61,62,63,64]. Apart from the use of Gln in TCA anaploresis, it is also an activator of the mTORC1 pathway that stimulates cell growth and proliferation. This is both through direct activation of mTORC1 and the facilitation of Leu uptake via LAT1, which is also a well-known activator of mTORC1, but through a different mechanism [65,66]. However, it has been demonstrated that ASCT2 can promote the growth of cancers independently of LAT1 [67]. The upregulation of ASCT2 seen in cancer is believed to be driven by the Myc proto-oncogenes that encode activation factors. c-Myc is well known to activate the transcription of Gln metabolism-related genes and has been shown to bind to the promoter region of ASCT2 and thereby increase its expression [17]. Similarly, N-Myc has been shown to drive an increased expression of ASCT2 in MYCN-amplified neuroblastoma cells, along with activating transcription factor 4 (ATF4) [68]. Loss of function of the tumor suppressor retinoblastoma (Rb) protein, which is seen in multiple cancers, also appears to increase ASCT2 expression through the E2F-3 transcription factor [69]. Gln appears to have a role in the expression of ASCT2, both through interaction with the promoter region as well as a posttranscriptional effect [70,71].

### 3.3. Inhibitors of ASCT2

Several of the inhibitors known to inhibit ASCT2 are based on amino acid scaffolds, as can be seen in Figure 2. The first identified inhibitors were benzylserine and benzylcysteine, which showed competitive inhibition as well as inhibiting the anion leak catalyzed by ASCT2. Benzylserine was based on the previously known inhibitor of EAATs threo-beta-benzyloxyaspartate (TBOA) by removing the negatively charged Asp side chain. However, these inhibitors had weak potencies with K_i_ values around 1 mM [72]. Furthermore, benzylserine appears to inhibit ASCT1, LAT1, LAT2, SNAT1, and SNAT2, making it a nonspecific inhibitor [22,73,74].

Following this, a group of N-aryl glutamine derivatives was tested for their ASCT2 inhibition, hypothesizing that more acidic hydrogen on the amide nitrogen would increase the affinity for the transporter. This led to the inhibitor L-γ-glutamyl-p-nitroanilide (GPNA), which is widely known as an ASCT2 inhibitor [75]. However, GPNA appears to also inhibit the LATs and SNATs [22,76].

In 2012, a series of serine esters were evaluated by first doing docking studies on an ASCT2 homology model based on the homologous archaeal Asp transporter Glt_Ph_. Both the docking and experimental studies supported the hypothesis that larger and more hydrophobic substituents in the serine ester side chain would transform the derivatives into inhibitors instead of substrates. This led to the new and, at this point, most potent inhibitor of ASCT2 serine biphenyl-4-carboxylate with an apparent affinity of 30 µM (Figure 2) [77]. In the same year, a series of 1,2,3-dithiazoles were tested for their inhibition of ASCT2 in proteoliposomes. These compounds were not based on an amino acid scaffold but on their ability to covalently bind Cys residues in ASCT2 (suggested to be Cys-207 or Cys-210). The six most potent compounds in this series had potencies in the 3.7–10 µM range [78].

Later, a structure-based drug design approach was used to discover that the proline derivative γ-2-fluorobenzyl proline (γ-FBP) also inhibits ASCT2 [79]. This was a rather interesting finding, given that Pro is not known to be a substrate or inhibitor of ASCT2. This led the same group to investigate the γ-benzyl proline as a novel scaffold for the development of ASCT2 inhibitors. γ-(4-biphenylmethyl)-L-proline (Figure 2) was the most potent inhibitor found in this series with a K_i_ of 3 µM in rat ASCT2, again highlighting the room for larger hydrophobic groups in the binding pocket [80].

In agreement with the trend of larger hydrophobic inhibitors, Schulte et al. discovered 2,4-diaminobutanoic acid as a novel scaffold, which resulted in a series of 2-amino-4-bis(aryloxybenzyl)aminobutanoic acids with the most potent (Compound **12**) inhibiting ^3^H-Gln uptake in HEK293 cells with an IC_50_ of 7.2 µM [81]. Later, they tested the anticancer effects of a similar compound, V-9302 (IC_50_ = 9.6 µM, Figure 2), which was discovered in the same series as compound **12**. Both in vitro and in vivo studies showed the antitumor capabilities of V-9302 [82]. However, the inhibition of ASCT2 by Compound **12** and V-9302 could not be replicated by Bröer et al., who instead showed inhibition of SNAT2 and LAT1 [83].

Phenylglycine was later explored as a scaffold to discover inhibitors of both ASCT1 and ASCT2 due to their regulation of extracellular D-Ser levels in the brain. The study revealed that L-3,4-difluorophenylglycine (Figure 2) was the most potent ASCT2 inhibitor (IC_50_ = 131 µM), while L-4-chlorophenylglycine was the most potent in regard to lowering L-Ser uptake in rat astrocytes [84]. The study also revealed that the L-amino acid backbone was important for recognition by the two ASCTs, given that substitution of the carboxyl, amine, and α-carbon, or homologation, led to a significant loss of affinity. However, it should be noted that phenylglycine and derivatives are also known to interact with system L transporters [85].

Garibsingh et al. used a homology model of ASCT2 based on the crystal structure of EAAT1 to discover a new inhibitor that was not based on the amino acid structure [86]. Docking a lead-like ZINC15 library and prioritizing compounds for in vitro testing based on chemotype novelty and predicted mode of binding led to the discovery of compound **10** (Figure 2), which reveals a novel type of ASCT2 inhibitor. Compound **10** was 18-fold more potent in inhibiting ^3^H-Gln uptake in SK-MEL-28 cells (IC_50_ = 97.16 µM) compared to GPNA, and due to its unique mode of binding could provide the framework for developing more selective ASCT2 inhibitors [86].

Later, a new scaffold was explored by linking an amino acid backbone to a hydrophobic group through a sulfonamide of the sulfonic acid linker. The most potent compound, when tested in rASCT2, was a derivative of 4-hydroxy proline with a sulfonic acid linker joining to a biphenyl group with a terminal fluorine (Figure 2) [87]. The compound resembles the inhibitor identified by Singh et al. and appeared to inhibit EAAT1, thus resulting in a non-specific inhibitor once again [80,87]. Continuing this work, Garibsingh and colleagues showed that using *cis*-4-hydroxy-L-proline as a backbone resulted in better affinity than the *trans* and *D* isomers. Changing the linker to an ester and keeping the biphenyl sidechain resulted in submicromolar affinity (K_i_ in hASCT2 = 0.86 µM). Their study also showed how molecular dynamic simulations could be used in discovering pharmacologically relevant conformations of the ASCT2 binding site [88].

The latest effort in probing the binding pocket of ASCT2 was through a hydroxyhomoserine scaffold. Interestingly, using this scaffold allowed the development of an inhibitor (Figure 2) with selectivity for ASCT2 over ASCT1 and the EAATs while retaining submicromolar affinity (K_i_ = 0.84 µM) [89]. Exploration of the SLC1 allosteric binding sites also led to the discovery of a novel allosteric ASCT2 inhibitor named compound **#302**. Despite lacking selectivity for ASCT2 over ASCT1 and having weaker affinity than the newest ASCT2 inhibitors (K_i_ = 238 µM), it still serves as a starting point for the development of other allosteric inhibitors [90].

**Figure 2 pharmaceutics-16-00197-f002:**
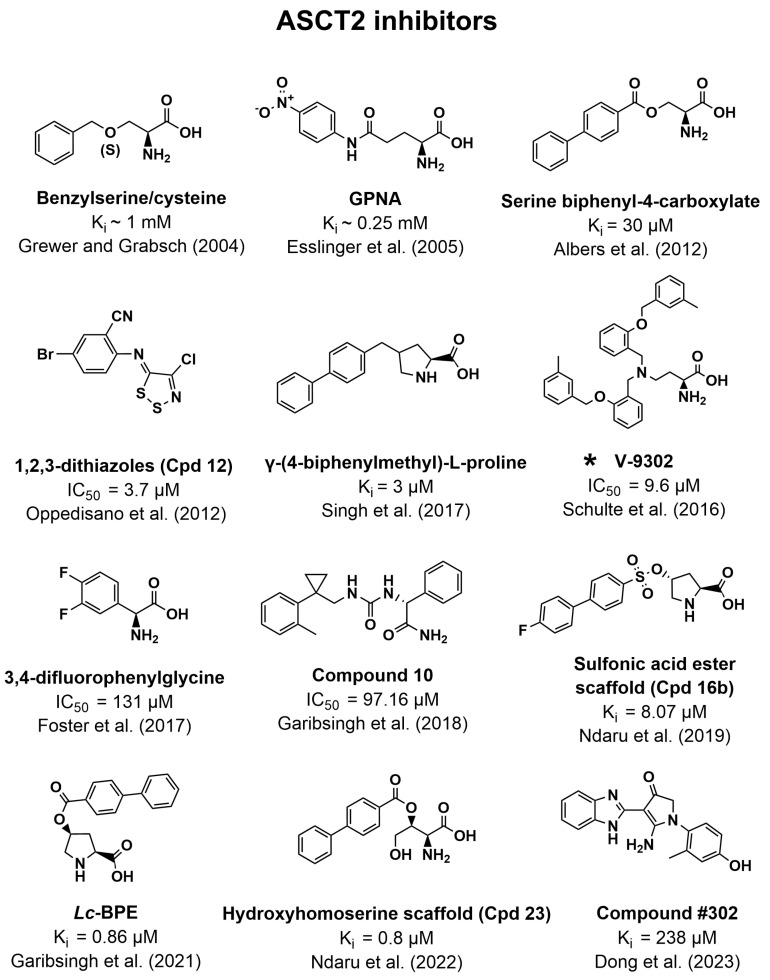
Overview of the structures of ASCT2 inhibitors and their inhibitory potency. References to where the inhibitory potencies were reported are given. * Indicates inhibitors with disputed inhibitory potency. For more information, see the main text [72,75,77,78,80,81,84,86,87,88,89,90].

## 4. Large Neutral Amino Acid Transporter 1 (LAT1)

### 4.1. Background

In 1998, the gene encoding for the large neutral amino acid transporter Lat1, *slc7a5*, was first cloned from rat C6 glioma cells, while the human LAT1 gene was cloned 1 year later [91,92]. The underlying protein appeared to have a substrate specificity similar to that of the system L described by Oxender and Christensen [91,93]. LAT1 facilitates the Na^+^ independent transport of bulkier branched-chain and aromatic amino acids such as Leu, Ile, Phe, Met, Tyr, His, Trp, and Val with K_m_ values in the 10–160 µM range and shows limited stereoselectivity [91,94,95]. It is also characterized by being inhibited by the system L-specific inhibitor 2-aminobicyclo-(2,2,1)-heptane-2-carboxylic acid (BCH) [91,96]. It was discovered that LAT1 also facilitates the transport of Gln and Asn, though with almost 100-fold poorer Michaelis constants when compared to the other substrates (K_m_~1.5–2 mM) [95]. The exchange of amino acids happens with a 1:1 stoichiometry, with the activity of the transporter being controlled by intracellular substrate concentrations due to a lower affinity at this site [97]. LAT1 is highly expressed in the brain, placenta, and skeletal muscle and is detectable in the heart, colon, thymus, spleen, kidney, liver, lung, and leukocytes [95]. LAT1 is also known to be highly expressed in the blood-brain barrier at both the apical and basolateral membranes and can be targeted by prodrugs to deliver them into the CNS [92,98,99]. LAT1 forms a heterodimeric complex through a disulfide bond with the glycoprotein 4F2 heavy chain (4F2hc, CD98, SLC3A2), which acts as a chaperone and helps traffic the protein to the cell membrane and stabilizes it. LAT1 is arranged into 12 transmembrane segments with the N- and C-terminal intracellularly, whereas 4F2hc forms a single transmembrane segment ([91,95,100] for review). The protein consists of 507 amino acids, and due to its shared identity with the Cationic amino acid transporters (CATs), it was placed in the SLC7 subfamily (SLC7A5) and is also part of the amino-acid-polyamine-organocation (APC) superfamily [91,92].

### 4.2. LAT1 Expression and Role in Cancer

Before being known as an amino acid transporter, a partial cDNA sequence of LAT1 was known as tumor-associated gene 1 (TA1) due to its differential expression in rat hepatoma cells [101]. This could be seen as an early hint of the importance of LAT1 in tumor development. The expression of LAT1 is increased in various cancers, such as clear cell renal cell carcinoma, colorectal cancer, endometroid carcinoma, gastric cancer, prostate cancer, esophageal cancer, NSCLC, and pancreatic cancer [102,103,104,105,106,107,108,109]. LAT1 also has a prognostic function in cancer, as its expression is linked to poorer survival rates [110,111,112]. It is believed that LAT1’s main contribution to cancer is by facilitating the uptake of essential amino acids, especially Leu, that in turn can activate mTORC1 [20]. For instance, it was shown that LAT1 was responsible for the majority of Leu uptake in T24 human bladder carcinoma cells [113]. Knocking down LAT1 using siRNA in various cancer cell lines led to significant growth inhibition, making it a possible therapeutic target in the development of cancer treatment [114]. Furthermore, LAT1 appears to be associated with the angiogenesis of tumor cells. It has been shown that along with its expression in cancer cells, LAT1 is also expressed in the surrounding vascular endothelium and linked to the pathological grade of the cancer [115]. This upregulation of LAT1 appears to be characteristic of tumor-associated vasculature and not that of normal tissue. Pro-angiogenic factors such as VEGF-A and FGF-2 seem to contribute to the induction of LAT1 in endothelial cells. Additionally, the role of LAT1 in angiogenesis also appears to revolve around mTORC1 signaling [116].

To further demonstrate the cancer-specific role of LAT1, the amino acid radiotracer L-3-^18^F-α-methyl tyrosine (^18^F-FAMT) is selectively taken up by LAT1 and thus accumulates in the tumor tissue [117]. Like GLUT1, LAT1 is used in cancer diagnosis, with ^18^F-FAMT showing higher specificity in detecting malignancies than ^18^F-FDG [118]. The analogous PET tracer 3-iodo-L-α-methyl-tyrosine (IMT) has also been shown to be selective for LAT1, and it is believed that this selectivity is due to the methyl group on the α-carbon found on both FAMT and IMT [117]. All in all, LAT1 has vast implications in cancer through various aspects such as diagnosis, prognosis, and treatment.

### 4.3. Inhibitors of LAT1

One of the first inhibitors used to distinguish system L activity as well as assess the impact of LAT1 inhibition on cancer growth is the bicyclic amino acid BCH (Figure 3) [96,113,119]. BCH has been shown to inhibit tumor growth both in vitro and in vivo and leads to decreased mTORC1 activity [120,121,122]. However, BCH is not selective towards LAT1 as it inhibits all the LAT transporters (LAT1-4) as well as ATB^0,+,^ and B^0^AT1 [123,124,125,126,127]. The affinity of BCH inhibition of LAT1 has been determined with a K_i_ of 156 µM [113].

The early structure-activity relationship (SAR) of LAT1 inhibitors was explored using Phe analogs. Here, it was shown that a free carboxylic acid group was necessary for recognition since dopamine, tyramine, and methyl esters of Phe had low affinity for LAT1 [128]. Introducing a methyl group to the amino group or changing the group to a hydrazine also led to decreased affinity. Hydrophobicity appeared to be of great importance for LAT1 recognition, and larger hydrophobic molecules like the thyroid hormones (triiodothyronine T3 (Figure 3), thyroxine T4) appeared to be functioning more as inhibitors (T3 K_i_ = 5.8 µM) than as substrates [128]. Later, SAR analysis also concluded that the carboxylic acid, amine, and hydrophobic side chain were important for binding to LAT1 [129]. However, they also revealed that the esterification of the carboxylic acid only decreased affinity slightly, indicating that the charge of the COOH group was not important for recognition. Furthermore, the distance between the carbonyl and alkoxy oxygen could be increased, as well as the distance between the carboxylic acid and amine with both β- and γ-amino acids having affinity [129,130].

Given that the natural aromatic and hydrophobic amino acids appeared to have the best affinity for LAT1, many of the discovered inhibitors are based on these scaffolds. In 2013, a virtual screening using homology models of LAT1 based on AdiC and ApcT led to the discovery of four new experimentally confirmed LAT1 inhibitors. Three of these were analogs of Phe/Tyr (3-iodo-L-tyrosine, 3,5-diiodo-L-tyrosine, and fenclonine), while the last one was a Gln analog (Acivicin) [37]. Acivicin (IC_50_ = 340 µM) and fenclonine were shown to be substrates of LAT1, while 3-iodo-L-tyrosine and 3,5-diiodo-L-tyrosine (IC_50_ = 7.9 µM) could only induce slight Leu efflux, indicating that they are better characterized as inhibitors. Furthermore, both acivicin and 3-iodo-L-tyrosine were shown to cause a LAT1-mediated inhibition of cancer cell proliferation [37]. Expanding on the Phe/Tyr scaffold, the SAR of meta-substitution was explored as it was hypothesized that this could fill a hydrophobic pocket in the LAT1 binding site. One of the main findings was that the introduction of larger lipophilic alkyl or aryl substituents at the meta-position decreased substrate activity but increased inhibitor activity. Phe with a phenyl group at the meta-position (Compound **29**, Figure 3) inhibited LAT1-mediated gabapentin uptake with an IC_50_ of 6.6 µM, rendering a likeness to the thyroid hormones in both structure and affinity [128,131]. Later, a novel series of LAT1 inhibitors based on the T3 structure was designed and studied [132]. The scaffold was based on an aminopropanoic acid group bound to a methoxy phenyl group with two chlorines at C-3 and C-5, while the methoxy group was bound to the naphthalene group with substitutions at the C-6 and C-7 position. Substitutions at the C-7 position appeared to be favorable, and the compound SKN103 with a 3-amino-phenyl group at the C-7 position resulted in an IC_50_ of 1.98 µM against LAT1 (Figure 3). SKN103 was able to significantly decrease the growth of cancer cell lines as well as enhance the antiproliferative effect of cisplatin [132]. Zur et al. looked at the LAT1 recognition of Phe/Tyr when replacing carboxylic acid with bioisosteres like tetrazole, acylsulfonamide, or hydroxamic acid [133]. Of these functional groups, only hydroxamic acid appeared to be a true bioisostere and retained LAT1 recognition despite having a pK_a_ of 6.9, which was furthest from the natural amino acid (pK_a_ = 1.8). These results made the authors hypothesize that the actual acidic functionality and corresponding pK_a_ of this group are less important than that of its hydrogen bonding capabilities [133].

In line with the trend of using amino acid scaffolds, it was discovered that Trp would make a good promoiety for designing prodrugs that target LAT1 [134]. Later, a tryptophan benzyloxy derivative (Compound **42**, Figure 3) was discovered as a LAT1 inhibitor (IC_50_ = 1.48 µM) as part of a virtual screen implementing both structure-based and pharmacophore approaches [135]. This hit was then later used as a scaffold but did not allow the discovery of significantly more potent LAT1 inhibitors [136].

Perhaps the most successful story in developing a LAT1 inhibitor based on an amino acid scaffold is that of KYT-0353/JPH203 (Figure 3). Being based on a Tyr backbone, JPH203 inhibited ^14^C-Leu uptake with a submicromolar potency (IC_50_ = 0.06–0.14 µM) and had an antiproliferative effect both in vitro and in vivo while showing great selectivity towards LAT1 over LAT2 [137]. The anticancer properties of JPH203 have been studied and confirmed in various cell lines such as colon cancer, oral cancer, thymic carcinoma, cholangiocarcinoma, osteosarcoma, thyroid carcinoma, renal cell carcinoma, medulloblastoma, gastrointestinal cancer, bladder carcinoma, pituitary tumor, lymphoma, and pancreatic cancer cells [137,138,139,140,141,142,143,144,145,146,147,148,149]. Phase 1 trials in 17 patients with advanced solid tumors showed tolerability and response at a dose level of 25 mg/m^2^ JPH203 given intravenously once daily for 7 days [150]. JPH203 is currently in phase 2 clinical trials for the treatment of advanced biliary tract cancer (UMIN000034080) and is thus a great example of the significance of targeting LAT1 in cancer treatment [150]. Furthermore, studies also indicate that JPH203 sensitizes cancer cells to radiation treatment, revealing a potential strategy for combination therapy [151].

Like ASCT2, LAT1 inhibition by 1,2,3-dithiazoles as well as 1,2,4-dithiazines was investigated. This scaffold also resulted in potent LAT1 inhibitors, with the most potent compound having K_i_’s of 0.76 µM (Compound **5**, Figure 3) and 1.13 µM (Compound **17**) [152]. The interaction was covalent binding to Cys residues, likely though mixed disulfide and trisulfide bonds, as seen for ASCT2 [78]. Homology modeling of LAT1 revealed interactions with the binding pocket and suggested that C407 was subject to covalent binding, which was confirmed experimentally [152]. The interaction was later investigated using molecular dynamics and density functional theory calculations to reveal important LAT1 residues that interacted with the inhibitors, as well as the crucial role of water in the covalent inhibition mechanism [153].

**Figure 3 pharmaceutics-16-00197-f003:**
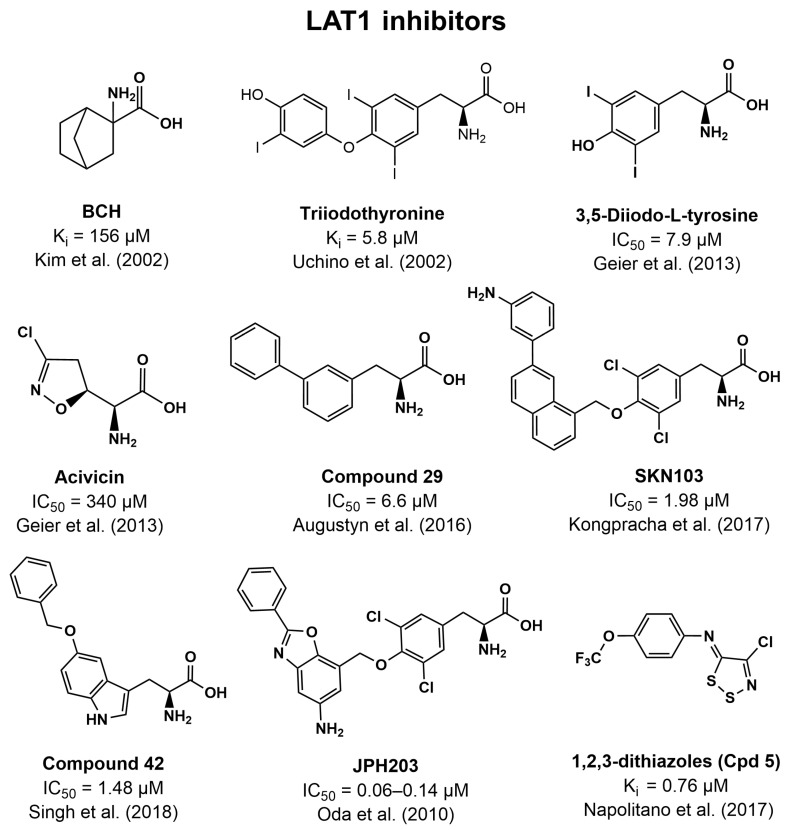
Overview of the structure of LAT1 inhibitors and their inhibitory potency. References to where the inhibitory potencies were reported are given [37,113,128,131,132,135,137,152].

## 5. Cystine/Glutamate Transporter (xCT)

### 5.1. Background

In 1980, Bannai and Kitamura studied the uptake of cystine and Glu into human diploid fibroblasts. This uptake system was sodium-independent, and the uptake of either amino acid could be inhibited by the presence of the other, suggesting a common carrier system [154]. It was later discovered that this carrier system functioned as an antiporter transporting cystine into the cell in exchange for a 1:1 ratio of Glu. With cystine acting like an anionic amino acid in this sodium-independent transport system, it was thus named x_c_^−^ [155,156]. The K_m_ values of the two substrates have been determined to be around 50 µM and 200 µM for cystine and Glu, respectively [154,155]. Both murine and human cDNA encoding the responsible transporter was discovered by Sato et al., and it became known that the x_c_^−^ system was due to a heteromeric transporter consisting of xCT (SLC7A11) and the 4F2 heavy chain (SLC3A2) [157,158]. xCT consists of 501 amino acids organized into 12 putative transmembrane domains, which was later confirmed when the structure was solved through cryo-EM [158,159]. The distribution of the transporter is tissue-specific, with the highest expression seen in the brain, epididymis, thyroid, and stomach [160]. It is also indicated that xCT expression is induced by immune activation in pathologies such as multiple sclerosis, which might serve as a link between inflammation and excitotoxicity [161]. xCT plays an important role in taking up cystine from the extracellular environment, which can be converted into cysteine once inside the cell. Extracellular cysteine is highly unstable and is thus mostly found in its disulfide form as cystine outside the cell. Long ago, it was suggested that cystine plays a crucial role in preserving cell survival, as evidenced by the inability of numerous cancer cell lines to endure cystine deprivation ([162,163] for review). It appeared that cystine was a major contributor to the formation of glutathione (GSH) and that cystine starvation led to cell death through oxidative stress [164]. This explains the role of xCT in maintaining redox homeostasis in the cell by providing it with sufficient cystine that can be converted to cysteine and combined with Gly and Glu to make GSH. The first step of GSH synthesis, where cysteine and Glu are combined, is considered the rate-limiting step, which emphasizes the important role of xCT in providing cysteine [165]. Most importantly, it was discovered that blocking the function of xCT led to a novel form of non-apoptotic cell death named ferroptosis [166]. This form of cell death is characterized by the accumulation of reactive oxygen species (ROS) in an iron-dependent manner since iron chelators can suppress ferroptosis [166]. Induction of ferroptosis has since become a popular topic as a mechanism for tumor suppression.

The expression of xCT is upregulated by various stimuli. Cystine deprivation has long been known to increase cystine uptake, but it has later been shown that deprivation of various amino acids leads to upregulation of xCT expression [167,168]. Furthermore, electrophilic agents such as diethyl maleate also induce xCT expression through electrophile response elements (EpRE) that are also found in genes encoding proteins that have defensive roles against xenobiotics and oxidative stress [169]. This again highlights the important role of xCT in the cell’s antioxidant defense.

### 5.2. xCT Expression and Role in Cancer

xCT has been shown to play an important role in cancer as xCT function and expression have been linked to prognosis in cancers such as acute myeloid leukemia, breast cancer, ovarian cancer, colorectal cancer, non-small cell lung cancer, prostate cancer, hepatocellular carcinoma, glioma, and melanoma [170,171,172,173,174,175,176,177,178]. The role of xCT in cancer is mainly to maintain redox homeostasis, as cancer cells are known to generate larger amounts of ROS due to their abnormal metabolism. xCT expression has also been linked to the *RAS* family, which is the most common mutated proto-oncogene in cancer. The transcription factor ETS-1 found downstream of the RAS-ERK pathway was shown to promote xCT expression, and it became known that xCT plays an important role in the KRAS transformation of tumor cells [179]. Interestingly, the tumor suppressor p53 represses the expression of xCT and implies that induction of ferroptosis is a natural form of tumor suppression [180]. Like its role in antioxidant defense, xCT also promotes resistance against chemotherapy. This chemoresistance has been shown in the case of cisplatin, gemcitabine, geldanamycin, and temozolomide and appears to be due to increased intracellular GSH levels [181,182,183,184].

xCT also appears to play a role in the altered metabolism of some cancers as its overexpression induces nutrient dependency. If cysteine or GSH is the net goal of xCT overexpression, the initial costs for this accumulation are Glu secretion and NADPH to convert cystine to cysteine [163]. Glu can be acquired from Gln through the action of glutaminase, and xCT can thus increase the Gln dependency of cancer cells [171,185]. xCT has also been linked to glucose dependency, which, of course, is another hallmark of the reprogrammed cancer metabolism [186]. The mechanism behind this glucose dependence is thought to be because of the need to reduce the less soluble cystine to the more soluble cysteine, which is accumulated in xCT overexpressing cells. As mentioned, this process uses NADPH, which can be supplied by the glucose-dependent pentose phosphate pathway [187]. This makes xCT overexpressing cancer cells susceptible to treatments that target glucose and Gln uptake.

### 5.3. Inhibitors of xCT

With the above examples of how xCT helps maintain redox balance and suppress ferroptosis in cancer cells, it is no surprise that targeting xCT with inhibitors has become a popular topic in cancer research. Early on, it was discovered that the anti-inflammatory drug sulfasalazine (Figure 4) is an inhibitor of xCT and was thus able to inhibit the growth of lymphomas [188]. This could be replicated in other cancers, such as prostate cancer and bladder cancer, while also increasing the cytotoxicity of cisplatin [189,190]. However, in a clinical trial, the treatment of glioma by sulfasalazine appeared to be unsuccessful [191]. Sulfasalazine has since been used as a scaffold in the search for new xCT inhibitors [192,193].

Like other amino acid transporters, analogs of the natural substrates of xCT were explored to uncover their inhibitory potential. It was found that the transporter preferred analogs with a side chain length like Glu or longer. Introduction of ring systems in the analogs also seemed to increase potency, as seen for ibotenate (K_i_ = 31 µM), (RS)-4-bromo-homoibotenate (K_i_ = 18 µM), L-quisqualate (K_i_ = 5 µM), and (S)-4-carboxy-phenylglycine (K_i_ = 5 µM), with the latter two characterizing more as true non-translocated inhibitors (Figure 4) [194].

Another commonly used xCT inhibitor is erastin, which was first discovered as part of a high-throughput screen looking for compounds that could selectively kill engineered tumorigenic cells [195]. In the discovery of ferroptosis, it was also discovered that erastin inhibits xCT and could thus explain part of its mechanism for antitumor activity by inducing ferroptosis [166]. It was also found that erastin was much more potent at inhibiting xCT (IC_50_ = 0.14–0.20 µM) compared to sulfasalazine (IC_50_ = 450–460 µM) [196]. SAR analysis of the erastin scaffold led to even more potent inhibitors with IC_50’_s down to 3.5 nM (Compound **21**, Figure 4) [196]. To make reversible covalent inhibitors, a series of ketone analogs of erastin were developed. Many of these analogs have increased potency compared to erastin while also showing increased solubility and metabolic stability, as the case for the compound termed IKE (imidazole ketone erastin, IC_50_ = 0.03 µM, Figure 4) [197].

Another approved drug, the kinase inhibitor sorafenib (Figure 4), was also shown to inhibit xCT and thus could induce ferroptosis in cancer cells. This property appeared to be unique when compared to other kinase inhibitors but might also be the reason for sorafenib’s significant association with adverse events [196]. However, it was later debated whether sorafenib was a genuine ferroptosis inducer, as this could not be replicated in a multitude of cancer cell lines [198]. A high-throughput screen discovered that the compound capsazepine, an antagonist of capsaicin, also inhibited xCT cystine uptake (IC_50_ of approximately 3 µM). Furthermore, it appeared that xCT inhibitors could have a role in reducing cancer-induced bone pain, which might, in part, be caused by the increased Glu secretion from xCT overexpression [199]. Another recent screen discovered 8 xCT inhibitor hits, most of these being based on a benzotriazole scaffold. The most potent inhibitor, HG106 (Figure 4), was able to decrease cystine uptake by about 50% at a concentration of 2.5 µM and showed selective cytotoxicity in *KRAS-mutated* cells [200].

**Figure 4 pharmaceutics-16-00197-f004:**
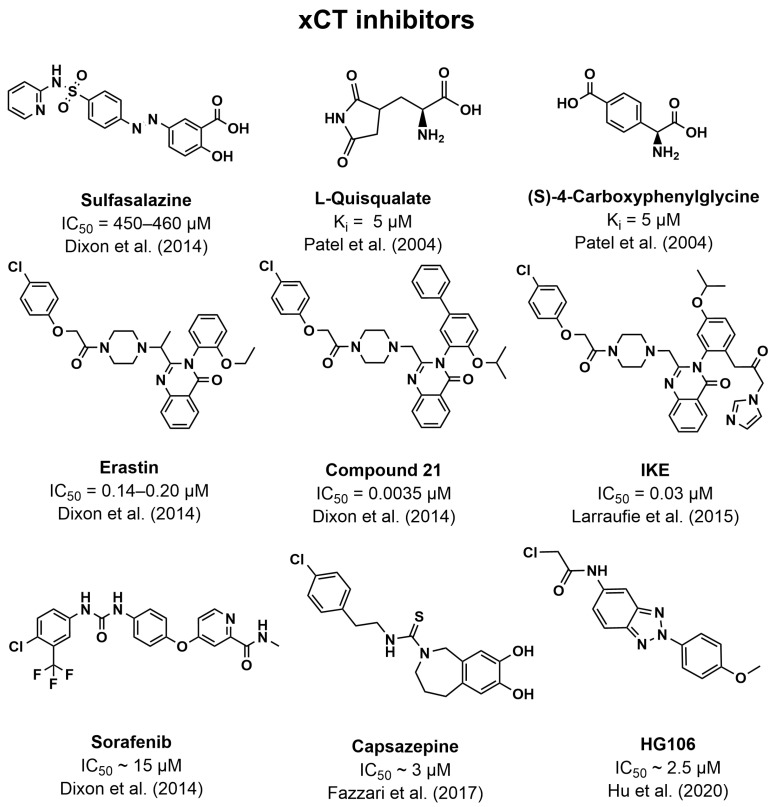
Overview of the structure of xCT inhibitors and their inhibitory potency. References to where the inhibitory potencies were reported are given [166,194,197,199,200].

## 6. Sodium-Coupled Neutral Amino Acid Transporter 1 and 2 (SNAT1/2)

### 6.1. Background

In 1965, it was discovered that a transport system facilitating sodium-dependent uptake of neutral amino acids could be inhibited by the amino acid analog α-methylamino-isobutyric acid (MeAIB) [201]. This system became known as system A and was later associated with proteins in the SLC38 family, namely SNAT1, SNAT2, and SNAT4 (previously named ATA1/2/3, respectively, for amino acid transporter A1/2/3). SNAT1 (SLC38A1) was cloned and characterized in 2000 [202], whereas SNAT2 (SLC38A2) was cloned soon after by three separate groups [203,204,205]. These transporters are known to transport smaller-medium aliphatic zwitterionic amino acids like Ala, Gln, Asn, Ser, Cys, Met, Gly, as well as His, which have K_m_ values of approximately 200–500 µM [205,206,207]. Pro is also transported but appears to be a better substrate for SNAT2 than for SNAT1 [205,207]. The uptake of these amino acids is coupled to a 1:1 symport of sodium and is highly sensitive to pH, with lower pH resulting in less uptake, likely due to a decreased affinity for sodium [205,206,207]. SNAT2 has also been shown to mediate an anion leak conductance that is increased by the presence of Na^+^ and inhibited by the presence of transported substrates [208]. SNAT1 is known to be expressed primarily in neuronal tissue, whereas SNAT2 is more ubiquitously expressed [202,204,205]. Hydropathy analysis of the protein sequences has predicted that the transporters contain 11 transmembrane domains with an intracellular N-terminus and extracellular C-terminus [209]. This membrane orientation was further supported by homology models using crystal structures of the bacterial proteins LeuT_Aa_ and Mhp1, as 3D database searches revealed relationships to the SLC38 families [210]. The SNAT1 and SNAT2 proteins contain 487 and 506 amino acids, respectively, and share approximately 50% of the sequence identity (using EMBOSS Needle) [211,212].

### 6.2. SNAT1/SNAT2 Expression and Role in Cancer

The activity of System A transport was discovered to be upregulated when cells were deprived of amino acids [213,214]. This adaptive regulation was later attributed to an increased expression of SNAT2, and it was shown that only system A amino acid substrates were able to suppress this effect [215]. This regulation has been connected to the enhancing effects of an amino acid response element (AARE) and a CAAT motif located in intron 1 of the *SNAT2* gene [216]. The phosphorylation of the α-subunit of eukaryotic initiation factor 2 (eIF2α) has also been shown to be needed for the induction of SNAT2 expression by amino acid starvation [217]. eIF2α is phosphorylated as part of the GCN2 pathway, which senses amino acid deprivation through the increase in uncharged tRNA. The induction of SNAT2 activity through AA starvation appears to be facilitated by both de novo synthesis of the protein as well as the recruitment from an intracellular pool of pre-made SNAT2 proteins [218]. Furthermore, the 5′ untranslated region of SNAT2 mRNA contains an internal ribosome entry site that ensures cap-independent translation of SNAT2 is fed as well as fasted conditions, which might otherwise decrease cap-dependent translation [217].

The activity of system A also appeared to increase in hyperosmotically treated cells [219,220]. The increase in activity appears to be dependent on de novo synthesis of SNAT2 [218]. This induction of SNAT2 seems to be faster than the other hyperosmotic stress-induced transporters such as TAUT, BGT1, and SMIT (as reviewed in [221]). The stimulation appears to be crucial for the recovery of cell volume after hypertonic induction of cell shrinkage, with the increased amino acid uptake resulting in a higher amount of intracellular osmolytes [222,223]. On the flip side, System A also appears to be involved in decreasing cell volume after hypotonic cell swelling due to a decrease in System A activity, resulting in less accumulation of amino acids intracellularly [221,224].

Given the importance of amino acids in cell metabolism and growth, it is hypothesized that some amino acid transporters play a key role in sensing and communicating amino acid availability. SNAT2 is thought to play such a role and has been described as a transceptor, indicating a dual functionality as both a transporter and receptor (as reviewed in [225]). For instance, it is shown that the SNAT2 substrate MeAIB, although being a non-metabolizable amino acid, is able to stimulate mTORC1 signaling [226]. This suggests that the act of substrate binding or translocation in SNAT2 might convey a signal that activates mTORC1. However, this mechanism has been mostly elusive, but recent crystal structures of SNAT9 (SLC38A9) in zebrafish appear to shed some light on how transceptors might function [227]. SNAT9 is a lysosomal transporter and functions as a luminal sensor of Arg and activator of mTORC1. The crystal structure shows that the N-terminus of SNAT9 is inserted in the Arg binding site during low luminal levels of Arg, suggesting a ball and chain model. Once Arg levels are higher, this N-plug will be released and facilitate binding to Rag GTPase complexes that, in turn, can activate mTORC1 [227]. Given that SNAT2 and SNAT9 belong to the same SLC family (SLC38), it seems possible that SNAT2 might have a similar mechanism involved in its transceptor function.

SNAT1 is highly upregulated in hepatocellular carcinoma when compared to normal liver tissue [228]. This upregulation was also seen in pre-cancerous liver tissue, and the knockdown of SNAT1 with siRNA lowered the cell viability in HepG2 cells, indicating a role for SNAT1 in the tumor growth and survival of hepatocellular carcinomas [228]. SNAT1 has also been shown to be upregulated in melanoma and plays a role in melanoma development through the promotion of cell proliferation and invasion [229]. Both SNAT1 and SNAT2 are highly expressed in breast cancer cell lines [230]. Knockdown of SNAT2 in these cell lines decreased Gln consumption as well as cell growth. Furthermore, high expression of SNAT2 was associated with poor survival in a large cohort of breast cancer patients, especially in the case of triple-receptor-negative breast cancer [230]. This indicates that SNAT2 could be a promising pharmacological target in breast cancer. In prostate cancer, sarcosine (the N-methyl derivative of Gly) is associated with the metastatic progression of the cancer and could be used to distinguish between benign and malignant cancer cells [231]. The accumulation of ^11^C-sarcosine in prostate cancer tissue of a single patient shows its potential use as a diagnostic tool as well as suggesting an upregulation of a solute carrier that facilitates this accumulation [13]. This was first believed to be the proton-coupled amino acid transporter PAT1, but it was recently shown that SNAT2 also plays a role in the uptake of sarcosine, at least in PC-3 cells [13,14].

### 6.3. Inhibitors of SNAT1/SNAT2

As mentioned previously, system A transport was first characterized by its ability to be inhibited by the substrate MeAIB (K_i_~0.3 mM, Figure 5), highlighting a unique role for system A in accepting N-methylated amino acids [201]. However, although often thought of as a unique system A inhibitor/substrate, MeAIB is also a substrate of PAT1 [232,233]. Nevertheless, MeAIB has been ubiquitously used as a non-metabolizable amino acid to inhibit system A activity due to a lack of other potent and selective inhibitors. Still, MeAIB cannot discriminate between SNAT1 and SNAT2 activity, whereas betaine has been shown to be a selective substrate for SNAT2 over SNAT1 [234].

Later, the inhibition of various Gln transporters by the Gln analogs L-γ-glutamyl-p-nitroanilide (GPNA, Figure 2) and L-γ-glutamylhydroxamate (GHX, Figure 5) and the Ser derivative benzylserine (BS, Figure 2) was studied. Here, it was shown that SNAT1 was inhibited by GPNA, GHX, and BS, whereas only GPNA and BS could significantly inhibit SNAT2 [22]. Later, it was shown that GHX inhibits SNAT2 with an IC_50_ of 0.33 mM [32]. However, these compounds could also inhibit ASCT2, SNAT4, and SNAT5, highlighting their lack of specificity [22]. Other nonspecific inhibitors include the 2-amino-4-bis(aryloxybenzyl)aminobutanoic acids compound **12** and V-9302 (Figure 2), first discovered to inhibit ASCT2, but also shown to inhibit LAT1 and SNAT2 [83].

Recently, a high-throughput screen of 33,934 compounds was used to find the potent inhibitor of SNAT2, 3-(N-methyl (4-methylphenyl)sulfonamido)-N-(2-trifluoromethylbenzyl)thiophene-2-carboxamide (MMTC, Figure 5). MMTC inhibited Pro uptake in amino acid-starved SKOV3 ovarian cancer cells with an IC_50_ of 0.8 µM [32]. The inhibitor appeared to be selective for SNAT2 over SNAT1 but did not appear to inhibit the growth of pancreatic, breast, and ovarian cancer cell lines within its solubility range. Yet, when combined with the GLUT1 inhibitor Bay-876, inhibition of cancer cell growth was seen, demonstrating a potential combined role of glycolysis and glutaminolysis in cancer cell growth [32]. However, the inhibition of SNAT2 by MMTC could not be replicated in hyperosmotically treated PC-3 prostate cancer cells. MMTC displayed limited solubility and could not significantly inhibit Gly uptake in PC-3 cells within its soluble range [235]. So, the potential use of MMTC as a SNAT2 inhibitor remains to be further investigated.

## 7. Proton-Coupled Amino Acid Transporter 1 (PAT1)

### 7.1. Background

The proton-coupled amino acid transporter, PAT1, transports neutral zwitterionic α-amino acids such as Pro, Ala, Gly, and γ-amino acids such as GABA with K_m_ values in the 2–20 mM range [232,236,237,238,239]. It also transports drug substances likely to be GABA-mimetics, such as gaboxadol, vigabatrin, and δ-aminolevulinic acid [240,241,242,243,244,245,246,247,248]. Substrate transport via Pat1 is coupled to proton transport and is rheogenic, resulting in transport activity under slightly acidic extracellular conditions and limited activity under conditions lacking a transmembrane proton gradient [249]. PAT1/Pat1 is under normal conditions, expressed in the luminal membrane of the intestinal epithelium and Caco-2 cells [233], in intracellular vesicles of mouse renal tubular cells [250], in lysosomes of rat neurons [251,252], and the nucleus of rat A7r5 smooth muscle cells [253]. The human PAT1 protein encoded by *SLC36A1* consists of 476 amino acids proposedly organized into 11 transmembrane segments with an intracellular N-terminus and extracellular C-terminus [254].

### 7.2. PAT1 Expression and Role in Cancer

PAT1 has been implicated in cell growth dependent on mTOR in a variety of experimental systems and could, therefore, have a role in cancer. The *Drosophila* PAT1 ortholog PATH is involved in cell growth of the eye related to mTOR [255]. When expressed in *Xenopus* oocytes, PATH appeared to be a high-affinity, low-capacity transporter and thus lacks the ability to markedly change intracellular amino acid availability. Thus, PATH can control cell growth in a way that does not require bulk transport of amino acids, which was the first suggestion that PATH and related PAT transporters can act as transceptors [255]. In human cells, Ögmundsdóttir et al. showed that PAT1 directly interacts with Rag GTPases in HEK293 cells and that these co-localized on late endosomes and lysosomes to promote activation of mTORC1-mediated growth [256]. This resembles the mechanism by which SNAT9 is believed to recruit and activate mTORC1 [227]. The phylogenetic relatedness of both the SLC36 and SLC38 families thus suggests that the transceptor function might be inherent to transporters within these families ([225,257] for review).

A study by Tuupanen et al. used exome-sequenced data from 25 sporadic microsatellite-instable colorectal cancers and searched for base-specific somatic mutation hotspots [258]. They found that the *SLC36A1* gene contains a hotspot mutation, c.406G>A, resulting in an Ala136 mutation [258]. This mutation was found in cancer with a low frequency [258]. Another study showed that *SLC36A1* carries mutations in primary colorectal tumors, and more than 10% of the investigated colorectal cell lines and *SLC36A1* mutations have also been reported in other primary tumors in the TCGA database [259,260]. On the other hand, *SLC36A1* has been identified as a reactive species-related gene associated with patient survival in Luminal B breast cancer through pathways regulating mitotic and mitochondrial functions [261]. In Gallbladder carcinoma (GBC), *SLC36A1* is the direct target gene of metastasis miRNA suppressors, miR-7-2-3p and miR-29c-3p. Moreover, the deficiency of miR-7-2-3p and miR-29c-3p was closely associated with poor prognosis of GBC patients [262]. This indirectly suggests that *SLC36A1* has a role in GBC progression, but the role remains unknown.

Overall, the role of the proton-coupled amino acid transporter SLC36A1 in cancer is still unclear; however, as it has been found to activate mTORC1 [256], it cannot be ruled out that *SLC36A1*/PAT1 has one or more roles in cancer metabolism and development. Furthermore, cancer cells are known to acidify their extracellular surroundings (as reviewed in [263]), which would increase the proton-driving force utilized by PAT1.

### 7.3. Inhibitors of PAT1

As for all solute carriers, one substrate may competitively inhibit the transport of another substrate, but some true inhibitors for PAT1 have also been identified. The first inhibitors were identified by the group of Matthias Brandsch, who surprisingly showed that amines and amino acids such as serotonin (K_i_ = 5.7 mM), L-Trp (K_i_ = 4.7 mM), and 5-hydroxy-tryptophan (K_i_ = 0.9 mM) were inhibitors of PAT1 (Figure 6) [264]. Later, Frølund et al. showed that dipeptides bearing -H as a side-chain such as Gly-Gly and Gly-Sar, similar to δ-aminolevulinic acid, were substrates of PAT1, while dipeptides such as Gly-Tyr (K_i_~20 mM), Gly-Pro (K_i_~16 mM, Figure 6), and Gly-Phe were inhibitors [265]. While the amino acid inhibitors had a similar affinity to the normal substrates, the dipeptides had a much lower affinity. Later, high-affinity inhibitors such as the anti-depressant sertraline (IC_50_ of 177–241 µM, Figure 6) [266] and 17-α-estradiol and 17-β-estradiol (E2, Figure 6) were identified [267]. Shan et al. suggested that estradiol binds to PAT1 and thereby changes the conformation of the PAT1 protein from an open to a closed state, thereby decreasing transport activity [267]. Recently, Nielsen et al. revisited this to assess the potential for drug-drug interactions, showing that 17-α-ethinyl-estradiol (E-E2, Figure 6) inhibited PAT1-mediated Pro and taurine transport in vitro in Caco-2 cells [268]. Under slightly acidic extracellular conditions, E2 and E-E2 inhibited Pro uptake with IC_50_-values of 10.0 ± 3.2 µM and 50.0 ± 14.3 µM, respectively [268]. The uptake of taurine was inhibited by E and E-E2 with IC_50_-values of 8.7 ± 5.1 µM and 22.9 ± 15.5 µM, respectively [268]. In contrast, the uptake of taurine via TAUT at neutral extracellular pH was not affected by E2 or E-E2 in the concentration range investigated [268]. The Pro-induced signal was attenuated when E2 and E-E2 were present in the Pro-containing solution being perfused over the oocyte. Buffers with E2 and E-E2 alone did not induce an inward current, showing that they were non-translocated inhibitors of PAT1 [268]. The physiological role of amino acids, dipeptides, and hormones acting as inhibitors of PAT1-mediated transport remains unknown.

## 8. Future Perspectives and Concluding Remarks

From the present review, it is clear that amino acid-transporting SLCs have gained interest in the pursuit of novel anticancer compounds. In addition to the transporters covered in the present review, a number of other amino acid transporters have gained an interest in cancer research. ATB^0,+^ (SLC6A14) transports all proteinogenic amino acids except the anionic ones and has been shown to be upregulated in a variety of different tumors (reviewed in [269]). Its potential as a therapeutic target in cancer has been shown in vitro and in vivo for ER-positive breast cancer and pancreatic cancer using the inhibitor α-methyltryptophan [270,271]. SNAT5 (SLC38A5) along with ATB^0,+^ have been shown to drive glutaminolysis and other cancer-relevant metabolic pathways (reviewed in [272]). SNAT5 is coupled to Na^+^ influx and H^+^ efflux and is thus able to alkalize cancer cells. It has recently been shown to be an important tumor promoter in pancreatic cancer [7]. The cationic amino acid transporter (CAT, SLC7A1-4, and SLC7A14) family has also gained interest in the field of cancer research, especially for their role in arginine transport and the importance of arginine in the tumor microenvironment [273].

However, the efforts and achievements in amino acid transporter research are a bit disproportionate, given that ASCT2 and LAT1 have multiple PDB structures (Table 1) and submicromolarly potent inhibitors available, whereas only a few inhibitors of SNAT1/2 and PAT1 have been reported they also lack structural information from, e.g., cryo-EM. Hopefully, with the development of structural biology techniques like cryo-EM [274] and structure prediction AI like AlphaFold2 [39], these transporters will soon have good quality 3D structures that can be explored through structure-based drug discovery, thereby further understanding their role and potential druggability in cancer. It is important to remember that 3D structures only serve as snapshots of the conformations a dynamic transporter undergoes during translocation. To fully understand the dynamics of transporters and help the efforts to chemically target them, multiple conformations of the transporter must be captured as crystal structures. Ironically, the availability of different potent inhibitors helps solve this problem as they can help lock the transporter in different conformations. This means that it is often a matter of getting a good crystal structure or a potent inhibitor to kickstart the process, which may explain the disparity between the development of, e.g., LAT1 inhibitors and SNAT2 inhibitors.

Another important point is that many inhibitors of amino acid transporters are based on substrate analogs (i.e., they resemble amino acids). This means that they are likely to bind to the same binding site as the substrates and thus compete with the natural substrates. This can be problematic given that, for example, amino acids like Ala and Gln reach plasma concentrations of about 500–600 µM [275] and thus might displace the inhibitor in vivo as a potential nutrient-drug interaction. Furthermore, given the abundance of amino acid transporters, it appears likely that amino acid-based inhibitors are non-selective and can bind to multiple amino acid transporters. This is, for instance, seen for a great deal of the ASCT2 inhibitors presented in this review. To alleviate this problem, efforts should be made to make non-competitive inhibitors. This could be either through allosteric inhibition or covalent inhibition. Allosteric sites can be elusive to target, given that most knowledge is based on endogenous substrates that bind to the orthosteric binding site. However, as previously mentioned, recent studies have helped elucidate how ASCT2 might be inhibited allosterically [90]. Covalent inhibitors overcome the competitiveness problem by leading to a prolonged inhibition of the amino acid transporter because of covalently binding to the protein. Often, the covalent binding is achieved by targeting reactive thiol groups of cysteine residues. However, cysteines are present in many proteins, so to help selectivity, targeted covalent inhibitor (TCI) strategies can be implemented. Here, the inhibitor is first guided to its target by non-covalent hydrophilic/hydrophobic interactions, and the proximity thus allows the nucleophilic thiol group to attack a weak electrophilic group (as reviewed in [276]). This methodology can be exemplified in the previously mentioned 1,2,3-dithiazole inhibitors that target ASCT2 and LAT1 [78,152].

Overall, amino acid transporters are a growing field within cancer research, with a multitude of transporters to explore and target. Many of the discussed approaches to discovering inhibitors of amino acid transporters can help inspire researchers on how to tackle other, less understood amino acid transporters and help the discovery of novel inhibitors.

## Figures and Tables

**Figure 1 pharmaceutics-16-00197-f001:**
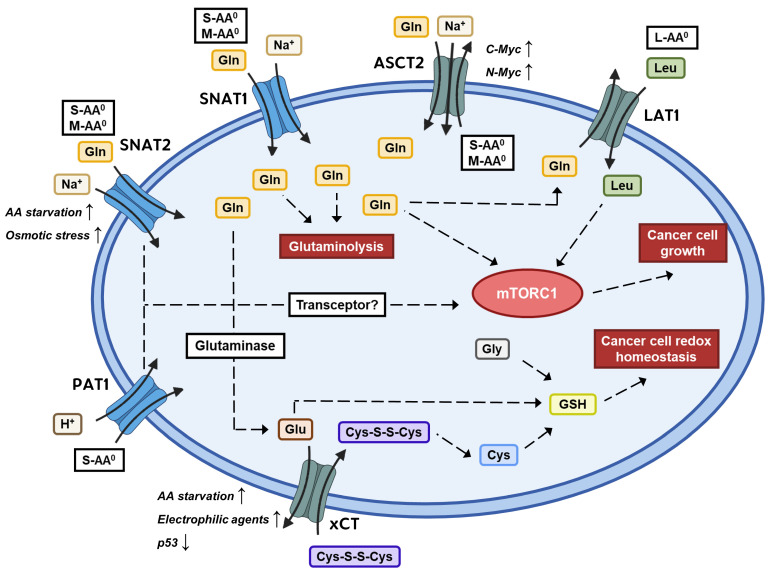
Overview of the roles and interconnection of select amino acid transporters in cancer growth. Potential determinants of cancer growth are highlighted in red. Factors that regulate the expression of the transporters are presented in *Italics* next to the transporter and indicate whether the expression is up- or downregulated. The figure draws inspiration from [25]. S-AA^0^: small neutral amino acids, M-AA^0^: medium neutral amino acids, L-AA^0^: large neutral amino acids, mTORC1: mammalian target of rapamycin complex 1, Cys-S-S-Cys: cystine, GSH: glutathione.

**Figure 5 pharmaceutics-16-00197-f005:**
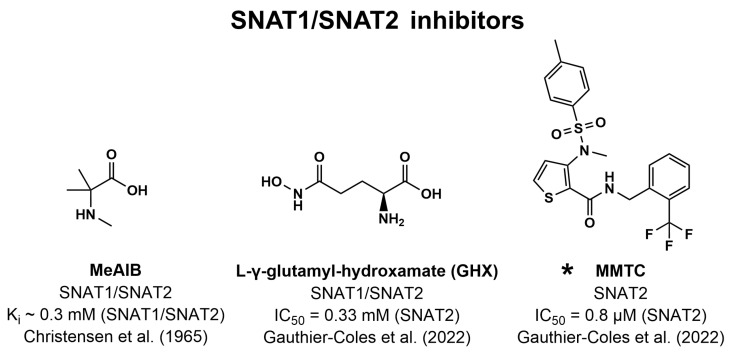
Overview of the structure of SNAT1 and SNAT2 inhibitors and their inhibitory potency. References to where the inhibitory potencies were reported are given. * Indicates inhibitors with disputed inhibitory potency. For more information see main text [32,201].

**Figure 6 pharmaceutics-16-00197-f006:**
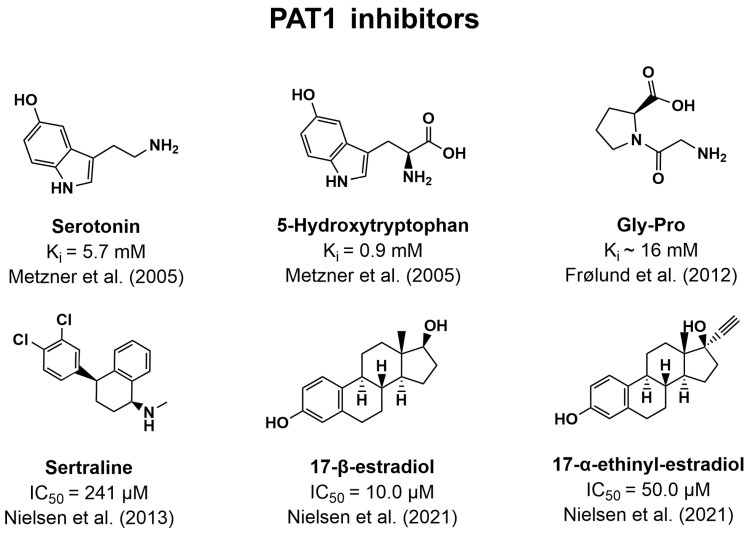
Overview of the structure of PAT1 inhibitors and their inhibitory potency. References to where the inhibitory potencies were reported are given [264,265,266,268].

**Table 1 pharmaceutics-16-00197-t001:** Overview of currently available PDB structures of ASCT2, LAT1, and xCT, along with their conformation, resolution, and ligand, if available. IF—inward facing, OF—outward facing.

Protein	PDB	Conformation	Resolution, Å	Ligand
ASCT2	6GCT	IF (occluded)	3.85	L-Gln
	6MP6	OF (open)	3.54	
	6MPB	OF (occluded)	3.84	L-Gln
	6RVX	IF (open)	3.61	
	6RVY	IF (open)	4.13	
	7BCQ	OF (open)	3.43	Lc-BPE (position “up”)
	7BCS	OF (open)	3.43	Lc-BPE (position “down”)
	7BCT	OF (open)	3.37	
LAT1-4F2hc	6IRS	IF (open)	3.30	
	6IRT	IF (open)	3.50	BCH
	6JMQ	IF (open)	3.31	
	7DSK	OF (occluded)	2.90	JX-075
	7DSL	OF (occluded)	2.90	JX-078
	7DSN	OF (occluded)	3.10	JX-119
	7DSQ	OF (open/occluded)	3.40	3,5-diiodo-L-tyrosine
xCT-4F2hc	7CCS	IF (open)	6.20	
	7EPZ	IF (open)	3.40	Erastin
	7P9U	IF (open)	3.70	L-Glu
	7P9V	IF (open)	3.40	

## Data Availability

Not applicable.

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
