# Peer review of "Exploring Amino Acid Transporters as Therapeutic Targets for Cancer: An Examination of Inhibitor Structures, Selectivity Issues, and Discovery Approaches"

_pharmaceutics, 2024, doi:10.3390/pharmaceutics16020197_

Round 1

Reviewer 1 Report

Comments and Suggestions for Authors

This review is focused on a new potential class of druggable targets for cancer: the SLCs that transport amino acids. The subject is fascinating, and the paper is informative and written.

The only concern is the first part of the section, "Discovering Amino Acid Transporter Inhibitors." Indeed, researchers interested in transporter inhibition do not need to learn about the meaning of Km, IC50, or inhibition mode; they probably know. I suggest avoiding those explanations and adding the details of the characteristics that ideally make a molecule a suitable inhibitor.

Reviewer 2 Report

Comments and Suggestions for Authors

This manuscript by Jakobsen and Nielsen provides a concise, logically structured, and highly informative overview of amino acid transporters known to be implicated in cancer. For each of these AA transporters, the authors provide historical background, outline the structural and functional characteristics, and discuss past and present efforts to develop specific and potent inhibitors. The review is backed by an impressive battery of 268 literature citations. The references encompass seminal original papers, other reviews and, where available, most recent advances. The paper is written in perfectly well-readable (albeit at times slightly repetitive) English. Overall, as a summary of our state of knowledge this review article is a useful contribution that might be of broad interest to academics working in the field. Without splitting hairs over every occasional typo (of which there are only few), I cannot add but very minor comments, which are as follows:

-          Line 47: typo: sodium / iodide (not iodine) symporter

-          Line 340: “resembles” is odd word usage in this context

-          Lines 384-6: Consider rearranging the sentence, e.g. by moving “through a disulfide bond” to before “with the glycoprotein”

-          Line 399: “thus” is redundant, as prognostic significance is not a necessary logical sequel of overexpression.

-          Lines 787-9: The message of sentence “As miRNAs […]” is unclear.

Reviewer 3 Report

Comments and Suggestions for Authors

The literature review by Jakobsen and Nielsen provides an overview of the major amino acid transporters implicated in cancer, their known substrates, and their various inhibitors. The amino acid transporters discussed are ASCT2 (SLC1A5), LAT1 (SLC7A5), xCT (SLC7A11), SNAT1 (SLC38A1), SNAT2 (SLC38A2), and PAT1 (SLC36A1). This review specifically focuses on the chemical structures of inhibitors for these transporters and discusses the issues in designing selective inhibitors. It also discusses different methodologies and approaches used to discover inhibitors. Overall, the manuscript is very well-written. It should contribute to the field, and hopefully propel the research forward.

Specific comments: 

1. Figure 1 – should show Na-dependency for ASCT2, and I also recommend horizontal text for ‘Glutaminase’

2. Change “an unspecific” to “a nonspecific” inhibitor (line 295)

3. Add units for IC50 (line 483). Also, it would be helpful to include all of the Ki’s or IC50’s in the text, rather than mentioning just some of them

4. Since PAT1 is a proton-coupled amino acid transporter, and pH is dysregulated in cancer, it would be useful to add a sentence or two providing this context and the implication for transporter function

5. Omit “known” (line 849)

6. There are additional amino acid transporters implicated in cancer that could be included here. If there are currently few to no inhibitors, then there could at least be some mention of these transporters in the Future Perspectives section.  For example, SLC6A14, SLC38A5, and SLC7A1/2

Reviewer 4 Report

Comments and Suggestions for Authors

The review article ‘Exploring Amino Acid Transporters as Therapeutic Targets for Cancer: An Examination of Inhibitor Structures, Selectivity Issues, and Discovery Approaches’ submitted by Sebastian Jakobsen and Carsten Uhd Nielsen, focuses on the inhibitors for amino acid transporters. Cells have evolved a variety of amino acid transporters to uptake the available amino acids from their surroundings. Cancer cells often exhibit elevated levels of these transporters to support rapid cellular growth, and studies have demonstrated that targeting these transporters by small molecules is detrimental to cancers. Understanding the available amino acid inhibitors and developing new ones is the need of time. The authors have beautifully explained the terminology of the transporters, which would be helpful for general readers.

Comments-

1.     Under the heading ‘Alanine, Serine, Cysteine transporter 2’, lines 224-226, the authors have used the terms ‘system L and A’ without addressing these terms.

2.     Under the heading ‘Large neutral amino acid transporter,’ lines 368-369, the authors mention ‘one year later’ but do not mention the year in the first clause of the sentence.

3.     Headings are necessary for different amino acid transporters.

4.     Correct the sentence in lines 321-322.
